# Influence of Nitrate Supplementation on Endurance Cyclic Sports Performance: A Systematic Review

**DOI:** 10.3390/nu12061796

**Published:** 2020-06-17

**Authors:** Jorge Lorenzo Calvo, Francesca Alorda-Capo, Helios Pareja-Galeano, Sergio L. Jiménez

**Affiliations:** 1Sport Department, Facultad de Ciencias de la Actividad Física y del Deporte, Universidad Politécnica de Madrid, 28040 Madrid, Spain; 2Faculty of Sport Sciences, Universidad Europea De Madrid, 28670 Madrid, Spain; xesca.alorda@gmail.com (F.A.-C.); helios.pareja@universidadeuropea.es (H.P.-G.)

**Keywords:** nitrate, nitric oxide, beetroot juice, performance exercise, endurance, cyclic

## Abstract

Endurance can be defined as the capacity to maintain one’s velocity or power output for the longest possible time. Maintaining such activity can lead to the onset of fatigue. Dietary nitrate supplementation produces an ergogenic effect due to the improvement of mitochondrial oxygen efficiency through a reduction in the oxygen cost of exercise that increases vasodilation and blood flow to the skeletal muscle in recreationally active subjects. However, the effects of dietary nitrate supplementation on well-trained endurance athletes remain unclear; such supplementation could affect more performance areas. In the present study, a systematic review of the literature was conducted to clarify the use and effects of nitrate as a dietary supplement in endurance athletes trained in cyclic sports (repetitive movement sports). A systematic search was carried out following the Preferred Reporting Items for Systematic Review and Meta-Analyses (PRISMA) guidelines in the databases of SCOPUS, Web of Science (WOS), Medline (PubMed), and Sport Discus from 1 January 2010 to 30 November 2019. Twenty-seven studies were included in the study. The methodological quality of the articles was assessed using the McMaster Critical Review Form. Statistically significant ergogenic results were obtained in 8 (29.63%) of the 27 studies investigated, with significant results obtained for cardiorespiratory parameters and performance measures. Improvement in exercise tolerance was obtained, which could help with exhaustion over time, while the improvement in exercise economics was not as clear. Additionally, the dose necessary for this ergogenic effect seems to have a direct relationship with the physical condition of the athlete. The acute dose is around 6–12.4 mmol/day of nitrate administered 2–3 h before the activity, with the same amount given as a chronic dose over 6–15 days. Further studies are required to understand the factors that affect the potential ergogenic impacts of nitrate on athletic performance among endurance athletes.

## 1. Introduction

Nitrate supplementation enhances nitric oxide (NO) bioavailability via the NO_3_–nitrite–NO pathway, which is involved in several physiological processes that could potentially improve skeletal muscle function [1,2]. NO endogenous synthesis occurs through at least two different physiological pathways: the NO synthase (NOS)-dependent and NOS-independent pathways [2]. L-arginine and L-citrulline are the main precursors of NO in the NOS-dependent pathway, while nitrate (NO_3_^−^) and nitrite (NO_2_^−^) are the main substrates that produce NO via the NOS-independent pathway (Figure 1).

In addition to the endogenous generation of NO through the NOS pathway, NO_3_^−^ and NO_2_^−^ reserves can also be increased exogenically through one’s diet. Approximately 80% of the nitrates in one’s diet come from the consumption of vegetables [3], mainly through green-leaf vegetables, such as lettuce, spinach, arugula, celery, watercress, and beet. Beet is one of the main nitrate sources since it contains 250 mg (4 mmol) of nitrate per 100 g of fresh weight (Table 1).

NO signaling affects multiple physiological mechanisms that could potentially enhance skeletal muscle. NO mediates smooth muscle relaxation, which promotes vasodilation and blood flow regulation and thereby improves oxygen delivery and mitochondrial respiration (Figure 1). This, in turn, enhances type II muscle fiber function [5,6,7]. This physiological facilitation suggests that supplementation with nitrates could have ergogenic effects on cardiorespiratory endurance [8].

The ergogenic effects of nitrate have been demonstrated in endurance and submaximal exercises to improve high-intensity endurance performance [8,9]. This has led the scientific community to increase the number of investigations that analyze the ergogenic potential of nitrates under different exercise conditions, including high to low intensity, long to short duration, continuous and intermittent, normoxia and hypoxia, and acute dose and chronic exposure, with different nitrate sources and doses [10,11,12,13]. Supplementation with nitrates, either with pure nitrate salts or foods rich in nitrates, has been proven to enhance the sporting performance of recreationally active young people [14,15,16,17], while nitrate supplementation has not yet been shown to improve performance among non-professional athletes [18].

Endurance cyclic sports require repetitive movements, usually lasting more than five minutes, through predominantly aerobic metabolism. When an endurance athlete takes part in high-intensity effort, various physiological factors limit their performance, such as the maximum oxygen uptake, first and second ventilatory threshold, and energy efficiency and economy [8]. At the same time, these factors can potentially be improved through NO signaling via nitrate supplementation.

The objective of this systematic review was to analyze the ergogenic effect of nitrate supplementation on well-trained cyclic sports athletes, as well as to study the optimal sources and doses of nitrates.

## 2. Materials and Methods

### 2.1. Eligibility Criteria

This systematic review focused on the effect of nitrate supplementation on endurance cyclic sports performance. This review followed the PICOS question model for framing the research question and defining the inclusion criteria [19]: P (population): “endurance and healthy trained athletes between 18 and 50 years old”; I (intervention): “supplementation with nitrate”; C (comparison): “same conditions with placebo or control group”; O (outcomes): “performance measures”; S (study design): “double- or single-blind design and randomized parallel or crossed.”

### 2.2. Literature Search

A structured search was carried out following the Preferred Reporting Items for Systematic Review and Meta-Analyses (PRISMA) guidelines [20] in the SCOPUS, Web of Science (WOS), Sport Discus, and Medline (PubMed) databases. The search included original articles with randomized controlled crossover or parallel designs in which the intake of nitrate administered before and/or after the exercise was compared with an identical placebo situation. Due to the increasing number of investigations over the last 10 years interested in using nitrate as a supplement in athletes, the search was bounded from 1 January 2010 to 30 November 2019. Search terms were established matching the Medical Subject Headings (MeSH), as follows: “nitrate,” “nitric oxide,” “beetroot juice,” “performance exercise,” and “endurance.” The connectors OR and AND were also used to filter the search.

### 2.3. Study Selection

The following inclusion criteria were applied to select studies: (I) articles depicting a well-designed experiment that included the ingestion of a dose of nitrate, or nitric oxide or beetroot juice before and/or during exercise in healthy trained athletes; (II) the study investigated endurance cyclic sports; (III) there was an identical experimental situation with or without the ingestion of a placebo in a normoxic condition; (IV) there was a double- or single-blind design and randomized parallel or crossed design; (V) crossed over studies that did not allow for a clean-up period of over 24 h were excluded; (VI) there was clear information on the administration of nitrate; (VII) nitrate was administered in the form of a beverage, gum, or pills; (VIII) at least one of the measured performance variables was changes (effort shown, maximum power, average power, time, distance, oxygen consumption, saturated oxygen, total work, or number of completed sprints among others); and (IX) the language was restricted to English. The following exclusion criteria were applied: (I) animal studies, (II) uncontrolled trials, (III) studies using non-standardized turmeric extracts or extracts of unknown nitrate content, (IV) studies performed on non-trained or injured/sick subjects, (V) studies investigating children and the elderly, and (VI) studies in non-specific sports or non-endurance sports.

### 2.4. Data Extraction

The following information was extracted from the selected articles: study source (authors and year of publication), participant characteristics (level of activity or sports discipline, experience and training load described by the authors, number of participants and sex), age, VO_2max_, supplementation protocol (source of nitrates and duration), nitrate dose, time from the last dose of nitrate, exercise protocol, main outcome (performance measures), and results.

### 2.5. Quality Assessment and Risk of Bias

The methodological quality of the articles, evaluated using McMaster’s Critical Review Form [21], ranged between 13 and 15 points, representing a minimum methodological quality of 86.6% and a maximum of 100%. Of the 27 studies, 3 achieved 13 items, representing 11.11% of the total number of studies, 3 achieved 14 items, representing the same percentage of 11.11%, and the rest of the studies (21) achieved 15 items, representing 77.78% of the total number of studies. All studies achieved a “very good” quality. No study was excluded because it did not reach the minimum quality threshold. The main deficiencies found in methodological quality were associated with items 13 and 14 of the questionnaire and comprised the report of clinical importance and one study had drop-outs. The objective of this evaluation was to determine and to compare the quality between the different study designs (Table 2).

## 3. Results

### 3.1. Study Selection

The initial research obtained through database searches provided 4341 records. Of these records, 3102 duplicates were removed. Of the 1239 remaining records, a total of 1067 were excluded after screening the titles and abstracts for eligibility. Ultimately, 27 studies were included in this review. The PRISMA flow chart was used to summarize the systematic research and selection process (Figure 2).

### 3.2. Characteristics of the Studies

A total of 323 participants were included in these 27 studies. The average sample included 12 subjects but seven studies assessed the effects of nitrate in larger group subjects (up to 26 participants). The study mode was carried out using eight subjects. All subjects in the review were defined by the authors as either trained or well-trained using the various criteria listed in Table 3. Among the 323 experimental subjects, only 39 (12.07%) were women. Moreover, female athletes only took part in 5 of the 27 studies selected, with only one study exclusively on trained women. Most of the studies were performed with populations in their 20s and 30s. Most focused on cycling, with 9 studies (33.33%) and 101 participants (31.27%), followed by combined cycling and triathlon studies, with 97 participants (30%) and 6 studies (22.2%). We also found other less represented sports, such as swimming (3), athletics (2), rowing (1), and cross-country skiing (2). For the ergogenic effect of nitrate supplementation, statistically significant results were obtained in eight (29.63%) of the studies that investigated cyclic endurance sports performance [22,23,26,33,35,44,45].

The main form of supplementation was beetroot, except for three studies that used sodium nitrate [18,23,40] and one that used potassium nitrate [42]. Most of the studies used mmol as a measuring unit to determine the value of the dose of nitrate intake. The average dose of nitrate was 8.7 mmol/day (minimum 4 mmol/day and a 19.5 mmol/day maximum). Both acute and chronic supplementation protocols were found in the reviewed studies. Of the studies, 40.74% used chronic supplementation and 51.85% used acute supplementation, applying equally high and low doses for both acute and chronic treatments, and 7.4% used both types of supplementation.

Studies on chronic treatment were performed over an average of 6 days, taking into account the fact that some treatments lasted 3 days, and others, such as those described by Balsalobre et al. [22], applied supplementation for 15 days. The last dose of nitrate before physical performance testing was not given at the same moment in all the studies. The intervals used for administering the dose before the performance test was between 1.5 and 4 h, with the most common being 2–3 h (88.89%). Only the study of Balsalobre et al. [22], which featured the longest chronic supplementation (15 days), began the incremental test until exhaustion 24 h after the last dose.

Athletes in this systematic review had a moderate-to-high level of cardiorespiratory fitness. Most of the studies determined the VO_2max_ of their athletes. The VO_2max_ values ranged from 42 mL/kg/min to 85 mL/kg/min, with the lowest values found in studies using male master athletes in swimming [43]. Other measures of performance used to determine the effects of nitrate supplementation were time, distance average, and maximum power. Moreover, except for some gastrointestinal pain, secondary effects were not reported, and all doses of supplementation were tolerated. For example, in the study by Hoon et al. [33], two athletes described mild gastrointestinal pain after consuming a nitrate-rich drink.

### 3.3. Risk of Bias

Most of the studies did not present a risk of bias. They were carried out using healthy and uninjured athletes via a randomized double-blind control test, leaving a sufficiently long clean-up period lasting from 4 to 7 days (except for some that did not use a crossover). Approximately 10% were done using simple blind tests. The placebo drinks, in most cases, had the same packaging, color, smell, and taste as the nitrate-rich drinks to reduce the risk of bias.

Most studies gave clear instructions to participants to refrain from caffeine and alcohol intake for at least 24 h before performance testing. Further, identical diets were used in the experiments such that diet would not influence the results. Details of nitrate-rich foods (green vegetables, beets, strawberries, grapes, and tea) and the importance of eliminating them from one’s diet during the period of experimentation were given. Most participants maintained a moderate level of training during the previous days, in addition to refraining from physical exercise during the last 24 h before testing.

### 3.4. Group and Subgroup Effects

In all studies, the concentrations of nitrate and plasma nitrite increased significantly (*p* < 0.05) after nitrate supplementation compared to the placebo. Among the 27 studies reviewed, about 29.63% provided results that were statistically significant for nitrate intake. As shown in Table 4, of the eight articles, four showed significant results for both respiratory parameters and measures of performance [22,23,43,44], seven studies give results for measures of performance [22,26,33,35,43,44,45] of which three did not influence respiratory parameters [33,35,45], and five significant studies gave results on respiratory parameters [22,23,26,43,44], of which only one did not influence the measures of performance [23].

Boorsma et al. [24] used the highest acute (13 mmol) and chronic (19.5 mmol) nitrate doses of all the studies. Nevertheless, the two experimental tests (the incremental test and 1500 m timed trial) provided no significant results. Bescós et al. [23] demonstrated a significant improvement in oxygen consumption during their tests, but in neither case did this improvement increase sporting performance. The other studies showed that nitrate supplementation did not increase VO_2_ or improve performance.

## 4. Discussion

### 4.1. Forms of Nitrate Supplementation

This systematic review demonstrates that the ergogenic effects of nitrate supplementation on endurance cyclic sports performance occur mainly through the intake of beetroot juice. However, the consumption of beetroot juice allows for a substantial nitrate load to be ingested quickly and easily before training or completion. Beetroot juice is considered an ergogenic supplement with strong scientific evidence, according to the ABCD classification of the Australian Sports Institute (AIS) [8]. In the included investigations, a placebo drink with an insignificant nitrate content was used. In addition to nitrate, beetroot also contains several active compounds that can influence physiological responses to exercise, including betaine, antioxidants, and polyphenols [35]. The use of beetroot depleted in nitrate as a placebo allowed the effects of nitrate to be isolated from those of other potentially ergogenic components.

Other studies that applied different forms of nitrate supplementation, such as potassium nitrate or sodium nitrate, did not obtain significant results. Thus, this form of supplementation may be less effective than beetroot juice. For example, a study conducted in 2016 compared the effect of beetroot juice and sodium nitrate on performance in trained athletes and discovered that the same concentration of nitrate produced a larger reduction of oxygen uptake in the experimental group that used beet juice [3], but the study did not highlight any factors related to sports performance.

Balsalobre et al. [22], after supplementing runners with a dose of 6.5 mmol/day of nitrate for 15 days, performed an incremental test until exhaustion on a treadmill (from low to high intensity), showing significant results in time until exhaustion (S: 1269 ± 53.6 vs. PLA: 1230 ± 73.5 s), which means that the supplemented athletes performed better. The most intense part of the test was used to determine the speed associated with VO_2max_.

Similarly, the incremental test with a cycloergometer performed in a study on cyclists by Bescós et al. [23] demonstrated better results with an acute dose of 11.8 mmol of nitrate compared to a placebo, but these results were not statistically significant (S: 416 ± 32 vs. PLA: 409 ± 27 s). The main difference between both studies was the supplementation protocol (beetroot juice for 15 days and sodium nitrate for a single day). As previously noted, among the 27 studies, 40.74% used a chronic dose and the other 51.85% used an acute dose, making it difficult to determine the option with the best effect. Boorsma et al. [24] and Nyakayiru et al. [40] were the only studies that investigated both types of supplementation and obtained similar results for both acute and chronic supplementation (both negative results). Similarly, McQuillan et al. [38] performed four time trials using different days of supplementation (3, 4, 6, and 7 days), and all of them gave negative results. On the other hand, Cermak conducted two studies [26,27] with triathletes and cyclists with a similar VO_2max_, dose of nitrate, time between the last intake, and time trial. The only differences were the days of supplementation (6/1 day) and the total time taken for the time trial (16/60 min). Cermak et al. [26] found a significant improvement in performance, while the other study did not obtain better results for the test time. This could be due to the form of supplementation or the type of test done. Notably, the eight most significant studies [22,23,26,33,35,43,44,45] examined five different sports with different doses of supplementation (see Table 3 and Table 4); therefore, the best mode of supplementation to obtain superior performance among athletes remains uncertain.

### 4.2. Pharmacokinetics of Nitrate

In humans, before absorption through the stomach wall, approximately 25% of the nitrate consumed is absorbed by the salivary glands and reduced to nitrite via the action of nitrate reductase enzymes from the facultative anaerobic bacteria on the dorsal surface of the tongue. Next, this ingested nitrite is reduced to NO in the acidic environment of the stomach but a substantial amount undergoes systematic circulation as nitrite. Plasma nitrate levels increase rapidly within 30 min after supplementation, reaching the peaks of nitrate and nitrite at 1–2 h. Two to three hours later, nitrite and nitrate gradually fall, returning to the baseline after approximately 24 h. The average life of plasma nitrate in humans is approximately 5 h, with a considerable decrease 4 h after ingestion.

All studies that obtained favorable ergogenic effects from nitrate performed their experimental tests between 2 and 3 h after intake. The studies that started their tests before 2 h improved their exercise capacity, while those that started after 3 h failed to improve their exercise capacity [24,25,32,34,37,39,40]. Interestingly, only one study analyzed the effects of the supplement 24 h after intake, obtaining significant results [25].

The nitrate and nitrite levels in plasma when fasting are 20–50,000 mmol/L and 100–500 mmol/L, respectively. Additionally, estimations of the average consumption of nitrate and nitrite in Europe are 31–185 mg/day, with 100% bioavailability of nitrate in the diet [4]. Thus, the doses used in nitrate studies are 4 to 12 times greater than the typical daily intake of nitrate in Europe. Studies like that of Bescós et al. [23] used an acute intake of 10 mg per kg of body weight of sodium nitrate, representing an approximate nitrate dose of 11.8 mmol. With these values, the participants went from having similar nitrate concentrations before intake (S: 30 ± 12, PLA: 28 ± 10 µM) to having an increase in plasma nitrate levels three hours after intake in the experimental group (250 ± 80 µM), with no change observed in the placebo group (29 ± 8 µM).

Amongst the protocols used during the studies, those that decided to refrain from the use of mouthwash or chewing gum stand out. These hygienic products alter the bioavailability of nitrite by killing the oral bacteria necessary for nitrate’s bioactivation. A cross-study designed with seven healthy volunteers examined the effects of antibacterial mouthwash on the nitrite levels in saliva and plasma measured after the intake of sodium nitrate (10 mg/kg) [49]. In the control situation, the levels of nitrate and nitrite in saliva and plasma increased long after the intake of nitrate. Similarly, the use of mouthwash 15 min before the intake of sodium nitrate did not have an effect on the nitrate accumulation in saliva and plasma but prevented its conversion to nitrite in saliva and attenuated the increase in plasma nitrite. The removal of some oral bacteria with mouthwash alters the bioavailability of nitrite in plasma by limiting the bioactivation of nitrate in the mouth.

The eight studies that obtained a performance improvement via nitrate supplementation prohibited the use of mouthwash among the athletes until the experimental tests were carried out. Some tests prohibited the use of mouthwash during the entire period of supplementation [22], while others prohibited it during the last 24 h before the experimental test or from the last intake until the end of the test. On the other hand, studies such as Bescós et al. [18] and Lane et al. [34], which did not see a positive effect in the performance tests after the intake of sodium nitrate and beetroot juice, did not inform the athletes of the need to refrain from oral antibacterial products, which could have caused a smaller increase in NO.

### 4.3. Exercise Type and Ergogenic Response

Bescós et al. [23], through the administration of large amounts of dietetic nitrate to competitive cyclists and triathletes, determined that even with no improvement in cardiorespiratory adaptation to low and medium exercise intensity (submaximal workloads corresponding to 2.0, 2.5, 3.0, and 3.5 W·kg^−1^ of body mass), the consumption of oxygen was much more efficiently reduced under high-intensity exercise. This highlights the importance of high-intensity exercise in the performance of cyclic sports.

The ergogenic effect of the beetroot juice was mediated by the influence of NO in the improvement of the mitochondrial oxygen efficiency, which increased vasodilation and muscular blood flow, among other factors related to exercise performance [50]. One of the effects produced by these physiological mechanisms is an improvement of the running economy, which translates into being able to run at the same pace with less effort or being able to run faster with the same energy consumption. In this systematic review, a total of 27 trials were carried out to verify the improvement of endurance through nitrate supplementation. Among them, seven trials showed significant improvement in the time of distance traveled [26,33,35,44,45]. In most of the studies that measured cyclists and triathletes with cycloergometers, different tests of 1 km up to 80 km were employed. Two studies applied a 10 km test and obtained a significant improvement in the time performance and average power when the athletes ingested the supplement within 3 h before starting the test [26,44]. It should be noted that the study by Rokkedal et al. [44] applied the same distance with a shorter time and lower oxygen consumption, which means that the athletes performed at a higher intensity.

On the other hand, Nyakayiru et al. [40] and Shannon et al. [45] did not find a significant improvement in time with the 10 km test, an acute dose, or a chronic 6-day trial. This could be due to the expected time between intake and the start of the test (≥3 h) since the ingested doses and VO_2max_ levels of the cyclists and triathletes in both studies were very similar. However, more studies are necessary to confirm this result.

Studies conducted with distances less than or greater than 10 km did not obtain any significant improvement in exercise capacity [18,25,27,28,31,32,37,38,39,40], except for a study where the yield was verified at 4 and 16.1 km with an acute dose of 6 mmol of beet juice and an improvement in time, and the average power was obtained [35]. This research, even using an acute intake, obtained highly beneficial results, with a time improvement of 2.7% in the 4 km race and 2.8% in the 16.1 km race. Meanwhile, acute doses of nitrate given to trained cyclists for cycling races of 80 km or more seemed to have no ergogenic effect [46]. These results suggest that, as distances become greater, the importance of intensity as a determining factor in performance decreases.

The only study on rowers showed a positive effect on the time required to complete the 2000 m test (via a remover gauge) when using a dose of 8.4 mmol of nitrate, while with a reduced dose (4.2 mmol of nitrate), this effect was not observed [33]. This study demonstrates the importance of applying a higher dose such that the NO level can have an ergogenic effect. Additionally, a 2000 m race is a test that rowers are highly accustomed to; they perform such a race monthly since it is part of the national rowing selection protocol.

The mechanism by which nitrates improve the efficiency of submaximal exercise remains contentious. It is believed that increasing NO levels after supplementation can reduce the cost of ATP production, thereby improving energy production efficiency [51]. This effect is believed to be related to a lower production cost of mitochondrial ATP [6] or muscle contraction ATP [52]. It is known that one of the costliest energy processes during skeletal muscle contractions is the pumping of calcium into the sarcoplasmic reticulum, which accounts for up to 50% of the total ATP turnover. In this way, the efficiency of oxidative phosphorylation is improved under a given relative intensity and the decomposition of phosphocreatine is reduced. In this systematic review, we found 10 investigations that evaluated whether an improvement in VO_2_ is observable using incremental tests at different levels of VO_2max_ or maximum power (W_max_). Four studies that investigated incremental tests until exhaustion obtained significant results for sports performance [22,43], except for one study that found significant results in respiratory parameters but no differences in the duration of incremental exercise between the experimental group and the placebo group [23]. This was the only study that used an acute dose, which could be one of the attributable causes. These findings show that, in athletes with a high level of training, nitrate supplementation may lead to a greater tolerance of submaximal effort, while at the same time casting doubt on the decrease in oxygen consumption. Other physiological parameters, such as VCO_2_ or mean heart rate, showed no significant differences between groups; neither did the perceived exertion. Only the study by Balsalobre et al. [22] showed significant improvements in the rating of perceived exertion (RPE).

A study by Pinna et al. [43] with trained male swimmers is a clear example of improved exercise efficiency. The training frequency of swimmers ranged from 3 to 4 times/week, with a 3000–5000 m distance covered each workout. With supplementation of 5.5 mmol of nitrate for six days, an incremental test was carried out until exhaustion in the pool using an elastic cord connected to a digital dynamometer tied on a belt to the subject to study the workloads. The results showed that, through the intake of beet juice, a greater workload and lower aerobic energy consumption were achieved, together with the same VO_2_.

### 4.4. Performance Level and Ergogenic Response

While some studies reported that dietary supplementation with NO donors induced benefits in exercise performance, others found no positive effects. In this sense, the training status of the subjects seemed to be a determining factor. Studies with healthy untrained or moderately trained subjects showed that NO donors could improve their tolerance to aerobic and anaerobic exercise [14,15,16,17]. However, for high-level athletes, who are more adapted to training and exercise, nitrate supplementation seemed to have little to no effect, at least when the doses were equal to those used on recreational athletes [53]. Additionally, some studies found that the basal plasma nitrite levels of elite athletes were higher than those of recreational athletes, showing that plasma nitrite is directly proportional to exercise capacity [54]. Therefore, one’s aerobic level could affect the ergogenic benefits induced by dietary nitrate supplementation, and higher doses may be necessary to obtain a benefit for elite athletes. Additionally, the same doses of nitrate among elite and recreational athletes were shown to produce the same increases in plasma nitrate concentrations [55]. The ingestion of a moderate amount of inorganic nitrate might not lead to higher performance because the increase in free radicals of the nitrate–nitrite–NO pathway in well-trained subjects engaging in high-intensity exercise enhances the function of the endogenous antioxidant system of these athletes; consequently, there is a greater reduction in these free radicals. On the other hand, because they are not as heavily trained, recreationally active subjects lack adaptations and may be more exposed to oxidative damage when performing high-intensity sports. Thus, increasing the bioavailability of NO through the intake of nitrate could be a benefit for such athletes. Additionally, the benefits of nitrate supplementation may also depend on individual genetics or the effects of training on muscle oxygenation, mitochondrial function, and muscle fiber type composition. It has been suggested that nitrate supplementation may be particularly effective in improving the physiological and functional responses in type II muscle fibers [56].

It is assumed, in addition to the greater adaptation that athletes have to high-intensity sports, that some athletes will benefit more than others and that there may be responders and non-responders in a given group. In the study by Bescós et al. [18], seven subjects showed a small increase (<30%) in plasma nitrite levels after dietary supplementation with nitrate compared to placebo. These subjects were methodologically classified as “low responders” or “high responders.” Although no statistical differences were found in the performance parameters (distance and output power) between the two groups, during the 40-min time trial after treatment, there was a small reduction in VO_2_ (≈1.7%), VCO_2_ (≈3.4%), and the relationship between VO_2_ and power (≈1.1%) in the group of high responders. Additionally, blood lactate concentrations showed a tendency to decrease at 3 min after exercise (*p* = 0.076) in most responding patients. Similarly, even without being able to demonstrate a significant effect in the 1500 m race, two runners responded positively to acute supplementation of 19.5 mmol nitrate and chronic supplementation of 13 mmol of nitrate over 8 days [24]. These subjects improved their results by 5.8 and 5.0 s through acute intake and by 7.0 and 0.5 s through chronic intake. Additionally, in a submaximal incremental test among trained triathletes, an improvement in distance and maintenance was observed (albeit without obtaining significant results), meaning that “the need to individually analyze the positive or negative response to this supplementation in the case of athletes trained may be the best option” [30].

## 5. Conclusions

The variety of results makes it difficult to draw clear conclusions about nitrate supplementation among endurance cyclic sports performance athletes. Although most of the studies showed no significant differences in the performance of the endurance-trained athletes after nitrate supplementation, some studies indicated that with the same maximal and mean power, the VO_2_ values decrease after nitrate supplementation. Furthermore, the time to exhaustion was delayed with nitrate supplementation in a race time trial between 5 and 30 min. Both results suggest enhanced performance.

The dose necessary for a significant effect remains unclear since some results used acute doses of 12 mmol of nitrate and others applied smaller doses (up to 6 mmol/day of nitrate). Finally, the best period for ingestion is 2–3 h before competing. Further investigations are required to confirm the findings of the present study.

## Figures and Tables

**Figure 1 nutrients-12-01796-f001:**
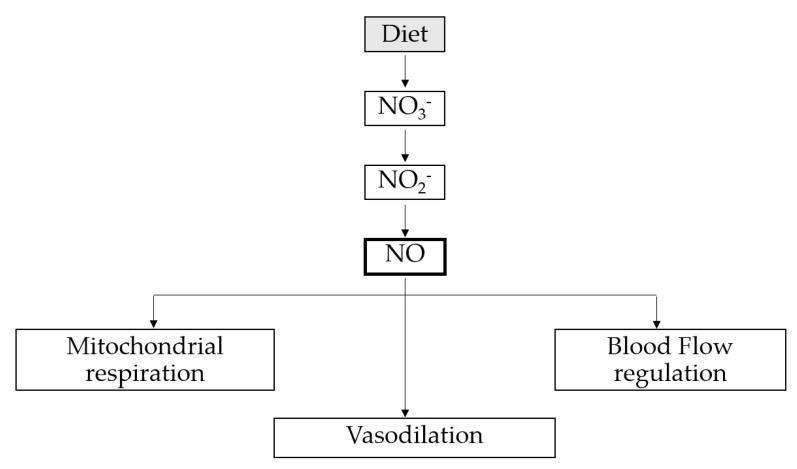
The main beneficial effects of nitric oxide on physical performance.

**Figure 2 nutrients-12-01796-f002:**
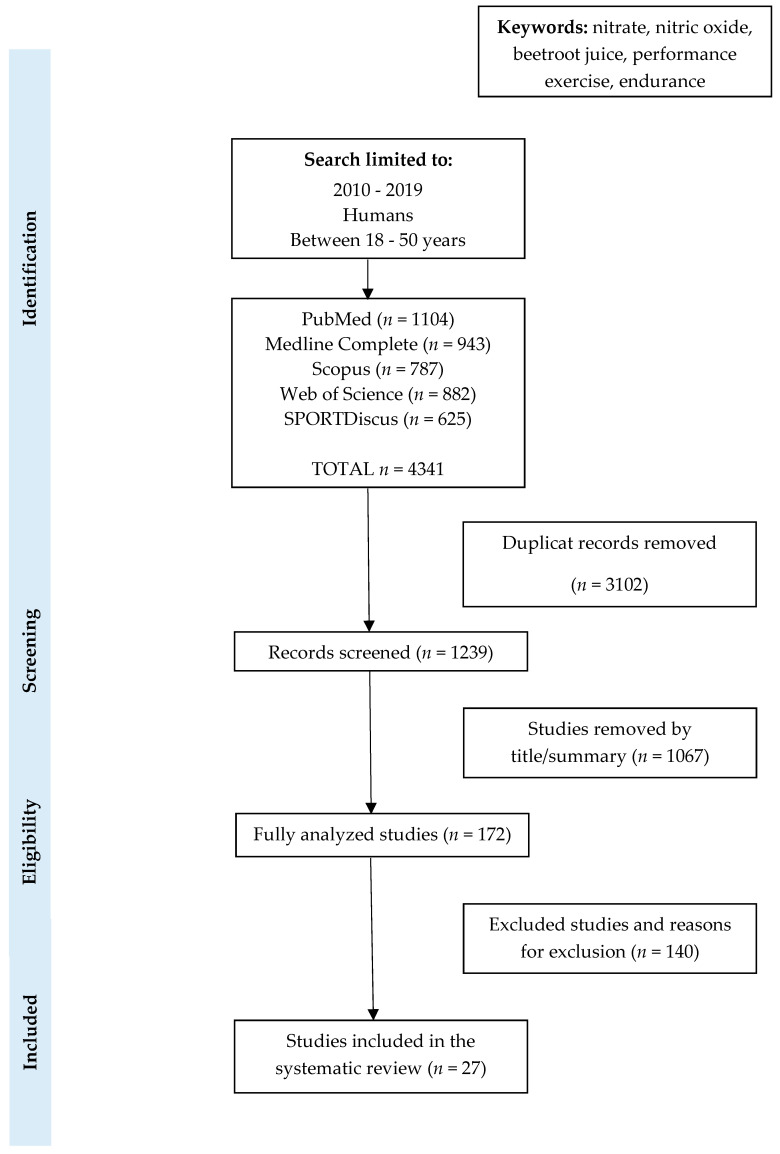
Preferred Reporting Items for Systematic Review and Meta-Analyses (PRISMA) flow. Summary of the systematic search and the study selection process.

**Table 1 nutrients-12-01796-t001:** Classification of vegetables according to nitrate content *.

Nitrate Content (mg/100 g Fresh Weight)	Vegetable Varieties
Very low, <20	Artichoke, asparagus, broad bean, eggplant, garlic, onion, green bean, mushroom, pea, pepper, potato, summer squash, sweet potato, tomato, watermelon
Low, 20 to <50	Broccoli, carrot, cauliflower, cucumber, pumpkin, chicory
Middle, 50 to <100	Cabbage, dill, turnip, savoy cabbage
High, 100 to <250	Celeriac, Chinese cabbage, endive, fennel, kohlrabi, leek, parsley
Very high, >250	Celery, cress, chervil, lettuce, red beetroot, spinach, rocket

* Information obtained from Hord et al. 2009 [4].

**Table 2 nutrients-12-01796-t002:** Methodological quality of the studies included in the systematic review.

Reference	1	2	3	4	5	6	7	8	9	10	11	12	13	14	15	Ts	%	MQ
Balsalobre et al. [22]	1	1	1	1	1	1	1	1	1	1	1	1	1	1	1	15	100	VG
Bescós et al. [18]	1	1	1	1	1	1	1	1	1	1	1	1	1	0	1	14	93.3	VG
Bescós et al. [23]	1	1	1	1	1	1	1	1	1	1	1	1	1	0	1	14	93.3	VG
Boorsma et al. [24]	1	1	1	1	1	1	1	1	1	1	1	1	1	1	1	15	100	VG
Callahan et al. [25]	1	1	1	1	1	1	1	1	1	1	1	1	0	0	1	13	86.6	VG
Cermak et al. [26]	1	1	1	1	1	1	1	1	1	1	1	1	1	1	1	15	100	VG
Cermak et al. [27]	1	1	1	1	1	1	1	1	1	1	1	1	0	0	1	13	86.6	VG
Christense et al. [28]	1	1	1	1	1	1	1	1	1	1	1	1	1	1	1	15	100	VG
Esen et al. [29]	1	1	1	1	1	1	1	1	1	1	1	1	1	1	1	15	100	VG
Garnacho et al. [30]	1	1	1	1	1	1	1	1	1	1	1	1	1	1	1	15	100	VG
Glaister et al. [31]	1	1	1	1	1	1	1	1	1	1	1	1	1	1	1	15	100	VG
Hoon et al. [32]	1	1	1	1	1	1	1	1	1	1	1	1	1	1	1	15	100	VG
Hoon et al. [33]	1	1	1	1	1	1	1	1	1	1	1	1	1	1	1	15	100	VG
Lane et al. [34]	1	1	1	1	1	1	1	1	1	1	1	1	1	0	1	14	93.3	VG
Lansley et al. [35]	1	1	1	1	1	1	1	1	1	1	1	1	1	1	1	15	100	VG
Lowings et al. [36]	1	1	1	1	1	1	1	1	1	1	1	1	1	1	1	15	100	VG
MacLeod et al. [37]	1	1	1	1	1	1	1	1	1	1	1	1	1	1	1	15	100	VG
McQuillan et al. [38]	1	1	1	1	1	1	1	1	1	1	1	1	1	1	1	15	100	VG
McQuillan et al. [39]	1	1	1	1	1	1	1	1	1	1	1	1	1	1	1	15	100	VG
Nyakayiru et al. [40]	1	1	1	1	1	1	1	1	1	1	1	1	1	1	1	15	100	VG
Nybäck et al. [41]	1	1	1	1	1	1	1	1	1	1	1	1	1	1	1	15	100	VG
Pawlak et al. [13]	1	1	1	0	1	1	1	1	0	1	1	1	1	1	1	13	86.6	VG
Peacock et al. [42]	1	1	1	1	1	1	1	1	1	1	1	1	1	1	1	15	100	VG
Pinna et al. [43]	1	1	1	1	1	1	1	1	1	1	1	1	1	1	1	15	100	VG
Rokkedal et al. [44]	1	1	1	1	1	1	1	1	1	1	1	1	1	1	1	15	100	VG
Shannon et al. [45]	1	1	1	1	1	1	1	1	1	1	1	1	1	1	1	15	100	VG
Wilkerson et al. [46]	1	1	1	1	1	1	1	1	1	1	1	1	1	1	1	15	100	VG

Ts—Total items fulfilled by study; 1—Criterion met; 0—Criterion not met; MQ—Methodological quality: P—poor ≤ 8 points, A—acceptable 9–10 points, G—good 11–12 points, VG—very good 13–15 points.

**Table 3 nutrients-12-01796-t003:** Summary of studies reviewed examining the effects of nitrate supplementation on trained athletes of cyclic endurance performance.

Reference	Participants	Age	Sports Experience and/or Training Load	VO_2max_ (mL/kg/min)	Supplementation Protocol	Nitrate Dose (mmol/day)	Last Dose (hours)	Exercise Protocol	Principal Performance Measures	Results
Balsalobre et al. [22]	M, middle and long-distance runners (*n* = 12)	26.3 *±* 5.1	International athletes with personal bests	71.8 *±* 5.2	BJ 70 mL (15 days)	6.5	24	Incremental running test to exhaustion	Time (s), RPE, SmO_2_ (%), VO_2_ (mL/kg/min)	Time (S: 1269 *±* 53.6 vs. PLA: 1230 *±* 73.5, D = yes), RPE (S: 6 *±* 1 vs. PLA: 7.7 *±* 1, D = yes), SmO_2_ (S: 31 *±* 6.9 vs. PLA: 27.7 *±* 4.8, D = yes), VO_2_ (S: 69.5 *±* 2.9 vs. PLA: 71.4 *±* 4.8, D = no)
Bescós et al. [18]	M, cyclists and triathletes (*n* = 13)	32.6 *±* 5.6	Experience in endurance events: 8 *±* 5 years, training volume: 15.7 *±* 5.0 h/wk	60 *±* 7	Sodium nitrate 250 mL (3 days)	11.6	3	Distance trial (40 min) in cycle ergometer	Distance covered (km), Mean power output (W), VO_2_ (mL/kg/min)	Distance covered (S: 26.4 *±* 1.1 vs. PLA: 26.3 *±* 1.2, D = no), mean power output (S: 258 *±* 28 vs. PLA: 257 *±* 28, D = no), VO_2_ (S: 51 *±* 7.9 vs. PLA: 50.9 *±* 6.6, D = no)
Bescós et al. [23]	M, cyclists and triathletes (*n* = 11)	34.3 *±* 4.8	Members of competitive squads	65.1 *±* 6.2	Sodium nitrate 250 mL (1 day)	11.8	3	Submaximal cycling trial (4 × 6 min) (T1), incremental exercise test to exhaustion (T2)	T1: VO_2_ (L/min) at 3.5 W/kg T2: time (s), maximal power (W), VO_2peak_ (L/min)	T1: VO_2_ (S: 0.5 *±* 0.11 vs. 0.53 *±* 0.06, D = no) T2: time (S: 416 *±* 32 vs. PLA: 409 *±* 27, D = no), maximal power (S: 416 *±* 29 vs. PLA: 410 *±* 28, D = no), VO_2peak_ (S: 4.64 *±* 0.35 vs. PLA: 4.82 *±* 0.33, D = yes)
Boorsma et al. [24]	M, distance runners (*n* = 8)	23.8 *±* 5	Elite distance; experience: provincial, national, or international events; training volume: 12.3 *±* 4 h/wk	80 *±* 5	BJ 210 mL (1 day)	19.5	1.5	Submaximal treadmill run (T1), time trial 1500 m (T2)	T1: VO_2_ (mL/min) T2: time (s)	T1: VO_2_ (S: 4192 *±* 113 vs. PLA: 4194 *±* 90, D = no) T2: time (S: 250 *±* 7 vs. PLA: 250 *±* 4, D = no)
					BJ 210 mL (on the test day) and 140 (other days) (8 days)	19.5 and 13	2.5	Submaximal treadmill run (T1), time trial 1500 m (T2)	T1: VO_2_ (mL/min) T2: time (s)	T1: VO_2_ (S: 4299 *±* 92 vs. PLA: 4216 *±* 95, D = no) T2: time (S: 250 *±* 5 vs. PLA: 251 *±* 4, D = no)
Callahan et al. [25]	M, cyclists (*n* = 8)	34 *±* 7	Well-trained cyclist	65.2 *±* 4.2	Beetroot crystals 15 g (3 days)	5	1	Time trial 4 km	Mean power output (W), time (s)	Mean power output (S: 388 *±* 54 vs. PLA: 386 *±* 56, D = no), time (S: 337.4 *±* 17.1 vs. PLA: 338.1 *±* 18, D = no)
Cermak et al. [26]	M, cyclists and triathletes (*n* = 12)	31 *±* 3	Training history of ≈10 years, training volume: ≈10 h/wk.	58 *±* 2	BJ 140 mL (6 days)	8	2.5	Submaximal cycling trail (2 × 30 min) (P1), time trial 10 km (T2)	T1: VO_2_ (L/min) T2: mean power output (W), Time (s)	T1: VO_2_ (S: 2.94 *±* 0.1 vs. PLA: 3.1 *±* 0.09, D = yes) T2: mean power output (S: 294 *±* 41.5 vs. PLA: 288 *±* 41.5, D = yes), time (S: 953 *±* 72.5 vs. PLA: 965 *±* 72.5, D = yes)
Cermak et al. [27]	M, cyclists and triathletes (*n* = 20)	26 *±* 1	Training experience: >3 times/wk for several years	60 *±* 1	BJ 140 mL (1 day)	8.7	2.5	Time trial ≈1 h	Mean power output (W), time (min)	Mean power output (S: 275 *±* 7 vs. PLA: 278 *±* 7, D = no), time (S: 65.5 *±* 1.1 vs. PLA: 65 *±* 1.1, D = no)
Christensen et al. [28]	M, cyclists (*n* = 10)	29 *±* 4	Experience: elite cyclists competing at the highest domestic level	72.1 *±* 4.5	BJ 500 mL (6 days)	8	3	Submaximal cycling trial at 70% W_max_ (2 × 6 min) (T1), repeated sprint test (6 × 20 s) (T2), time trial 400 kcal (≈20 min) (T3)	T1: VO_2_ (mL/min) T2: mean power output (W) T3: mean power output (W), time (min)	P1: VO_2_ (S: 792 *±* 82 vs. PLA: 806 *±* 82, D = no) P2: mean power output (S: 630 *±* 84 vs. PLA: 630 *±* 92, D = no) P3: mean power output (S: 290 *±* 43 vs. PLA: 285 *±* 44, D = no), time (S: 18.20 vs. PLA: 18.37, D = no)
Esen et al. [29]	M (*n* = 5) and F (*n* = 5), swimmers (*n* = 10)	22 *±* 6	Experience: ≥10 years competing at club standard and ≥5 years competing in regional and university-level. Training volume: 3–4 times/wk and 6–8 h/wk	2212	BJ 140 mL (3 days)	8	3	Time trial 200 m (T1), time trial 100 m (T2)	T1: time (s) T2: time (s)	T1: time (S: 152.6 *±* 14.1 vs. PLA: 152.5 *±* 14.1, D = no) T2: time (S: 69.5 *±* 7.2 vs. PLA: 69.4 *±* 7.4, D = no)
Garnacho et al. [30]	M, triathletes (*n* = 12)	39.3 *±* 7.5	Experience: National (*n* = 8) and international (*n* = 4) level of competition; training volume: ≥4 times/wk with ≥1 h/session	54.8 *±* 3.1	BJ 70 mL (1 day)	6.5	3	Time trial at VT1 (30 min) and VT2 (15 min)	VO_2_ (L/min), VCO_2_ (L/min), time (s) at VT2	VO_2_ (S: 73.8 *±* 8.9 vs. PLA: 72.2 *±* 9.4, D = no), VCO_2_ (S: 3.1 *±* 0.3 vs. PLA: 2.9 *±* 0.4, D = yes), time VT2 (S: 933 vs. PLA: 882, D = no)
Glaister et al. [31]	F, cyclists and triathletes (*n* = 14)	31 *±* 7	Experience: ≈13 years actively in sport; training volume: 10.7 *±* 2.2 h/wk at the time of the investigation	52.3 *±* 4.9	BJ 70 mL (1 day)	7.3	2.5	Time trial 20 km	Time (min)	Time (S: 35.33 *±* 1.5 vs. PLA: 35.37 *±* 1.7, D = no)
Hoon et al. [32]	M, cyclists (*n* = 26)	20.3 *±* 1.4	Trained male cyclists that were involved in a 6 week training camp at the Australian Institute of Sport	-	BJ 70 mL (1 day)	4.1	1.25	Time trail 4 min ×2	Mean power output (W)	Mean power output (S: 403 *±* 52 vs. PLA: 396 *±* 57, D = no)
							2.5	Time trail 4 min ×2	Mean power output (W)	Mean power output (S: 402 *±* 47 vs. PLA: 396 *±* 57, D = no)
Hoon et al. [33]	M, rowers (*n* = 10)	20.6 *±* 2.5	Highly trained; 2000-m personal-best time: 6 min, 17 s *±* 10 s; training volume: 16.9 ± 3.4 h/wk	-	BJ 70 mL (1 day)	4.2	2	Time trial 2000 m	Time (s)	Time (S: 383.4 *±* 8.7 vs. PLA: 3835 *±* 9, D = no)
					BJ 140 mL (1 day)	8.4	2	Time trial 2000 m	Time (s)	Time (S: 381.9 *±* 9 vs. PLA: 383.5 *±* 9, D = yes)
Lane et al. [34]	M (*n* = 12) and F (*n* = 12), cyclists and triathletes (*n* = 24)	31 *±* 7	Competitive level	71.6 *±* 4.6	BJ 140 mL (1 day)	8.4	2	Time trial 43.83 km	Time (min), power output (W)	Time (S: 64 *±* 2.8 vs. PLA: 63.5 *±* 3.2, D = no), power output (S:298 *±* 35 vs. PLA: 303 *±* 41, D = no)
		28 *±* 6	Competitive level	59.9 *±* 5.1	BJ 140 mL (1 day)	8.4	2	Time trial 29.35 km	Time (min), power output (W)	Time (*p* > 0.05), power output (S: 207 *±* 31 vs. PLA: 207 *±* 29, D = no)
Lansley et al. [35]	M, cyclists (*n* = 9)	21 *±* 4	Competitive level	56 *±* 5.7	BJ 500 mL (1 day)	6.2	2.5	Time trial 4 km	Time (min), mean power output (W), VO_2_ (L/min)	Time (S: 6.27 *±* 0.35 vs. PLA: 6.45 *±* 0.42, D = yes), mean power output (S: 292 *±* 44 vs. PLA: 279 *±* 51, D = yes), VO_2_ (S: 4.46 *±* 0.5 vs. PLA: 4.36 *±* 0.47, D = no)
								Time trial 16.1 km	Time (min), mean power output (W), VO_2_ (L/min)	Time (S: 26.9 *±* 1.8 vs. PLA: 27.7 *±* 2.1, D = yes), mean power output (S: 247 *±* 44 vs. PLA: 233 *±* 43, D = yes), VO_2_ (S: 4.32 *±* 0.47 vs. PLA: 4.19 *±* 0.56, D = no)
Lowings et al. [36]	M (*n* = 5) and F (*n* = 5), swimmers (*n* = 10)	20 *±* 1	Competitive level; training volume: ≥3 times/wk	−	BJ 140 mL (1 day)	12.5	3	Time trial 168 m	Time (s)	Time (*p* = 0.144), nitric oxide bioavailability +
MacLeod et al. [37]	M, cyclists (*n* = 11)	29.3 *±* 5.1	Trained cyclists that met the inclusion criterion VO_2peak_ > 5 L/min	67.5 *±* 5.8	BJ 70 mL (1 day)	6.5	2	Time trial 15 km	Time (s), mean power output (W/kg)	Time (S: 961 *±* 54 vs. PLA: 954 *±* 47, D = no), mean power output (S: 3.7 *±* 0.4 vs. PLA: 3.8 *±* 0.3, D = no)
McQuillan et al. [38]	M, cyclists (*n* = 9)	27 *±* 9	Endurance trained cyclists and competing in cycle races in the 3 months preceding the study; training volume: regular 11.4 *±* 2.6 hr/wk and 310.27 km/wk	68 *±* 3	BJ 140 mL (3 days)	8	2.5	Time trial 4 km	Time (s), mean power output (W)	Time (S: 341 *±* 12 vs. PLA: 340 *±* 10, D = no), mean power output (S: 390 *±* 45 vs. PLA: 393 *±* 37, D = no)
					BJ 140 mL (4 days)	8	2.5	Time trial 1 km	Time (s), mean power output (W)	Time (S: 79.6 *±* 3.5 vs. PLA: 79.2 *±* 2.9, D = no), mean power output (S: 8495 *±* 61 vs. PLA: 503 *±* 51, D = no)
					BJ 140 mL (6 days)	8	2.5	Time trial 4 km	Time (s), mean power output (W)	Time (S: 340 *±* 10 vs. PLA: 340 *±* 11, D = no), mean power output (S: 394 *±* 38 vs. PLA: 393 *±* 37, D = no)
					BJ 140 mL (7 days)	8	2.5	Time trial 1 km	Time (s), mean power output (W)	Time (S: 79.3 *±* 3.3 vs. PLA: 79 *±* 3, D = no), mean power output (S: 501 *±* 59 vs. PLA: 505 *±* 52, D = no)
McQuillan et al. [39]	M, cyclists (*n* = 8)	26 *±* 8	Well-trained endurance cyclists; training volume: weekly training duration 9 *±* 3 h/wk	63 *±* 4	BJ 70 mL (8 days)	4	2	Time trial 4 km	Time (s), mean power output (W)	Time (S: 343.6 *±* 14.3 vs. PLA: 344.8 *±* 14, D = no), mean power output (S: 380 *±* 41 vs. PLA: 375 *±* 40, D = no)
Nyakayiru et al. [40]	M, cyclists and triathletes (*n* = 17)	25 *±* 4	Competitive cyclist; experience: 9.6 *±* 5.1 years; training volume: 9.7 *±* 3.7 h/wk	65 *±* 4	Sodium nitrate 1097 mg (1 day)	12.9	4	Submaximal cycling trial at 45% W_max_ (30 min) and 65% W_max_ (30 min) (T1), time trial 10km (T2),	T1: VO_2_ T2: time (s)	T1: VO_2_ (D = no) T2: time (S: 1022 *±* 72 vs. PLA: 1017 *±* 71, D = no)
					Sodium nitrate 1097 mg (6 days)	12.9	4	submaximal cycling trial at 45% W_max_ (30 min) and 65% W_max_ (30min) (T1), time trial 10km (T2)	T1: VO_2_ T2: time (s)	T1: VO_2_ (D = no) T2: time (S: 1004 *±* 61 vs. PLA: 1017 *±* 71, D = no)
Nybäck et al. [41]	M (*n* = 5) and F (*n* = 3), cross-country skiers (*n* = 8)	21.8 *±* 2.8	Well-trained, competing at a national level	71.5 *±* 4.7 (M)	BJ 140 mL (1 day)	13	2.5	Submaximal test (2 × 6 min) (T1), time trial 1 km (T2)	T1: VO_2_ (L/min) T2: time (s)	T1: VO_2_ (S: 2.92 *±* 0.48 vs. PLA: 2.9 *±* 0.49, D = no) T2: time (S: 297 *±* 29 vs. PLA: 295 *±* 29, D = no)
Pawlak et al. [13]	M, triathletes and runners (*n* = 9)	21.7 *±* 3.7	Runners from clubs engaged in intense endurance exercise training and competition; inclusion based on VO_2max_ > 65 mL/kg/min	71.1 *±* 5.2	BJ 500 mL (3 days)	5.4	3	Repeated sprint test to exhaustion	No. of sprints completed, mean power output (W), VO_2_ (mL/min)	No. of sprints completed (S: 13.9 *±* 4 vs. PLA: 14.2 *±* 4.5, D = no), mean power output (S: 579.2 *±* 57.7 vs. PLA: 578.9 *±* 54.3, D = no), VO_2_ (S: 3378.5 *±* 681.8 vs. PLA: 3466.1 *±* 505.3, D = no)
Peacock et al. [42]	M, cross-country skiers (*n* = 10)	18	Experience: national and international standard junior skiers, all among the 20 best in the 2010 Norwegian Cup Series; training history: 502 ± 45 h/year	69.6 *±* 5.1	Potassium nitrate 1 g (1 day)	9.9	2.5	Submaximal test at 55% VO_2max_ (5 min) and 75% VO_2max_ (5 min) (T1), time trial 5 km (T2)	T1: VO_2_ (L/min) T2: time (s)	T1: VO_2_ (S: 3.77 *±* 0.62 vs. PLA: 3.89 *±* 0.39, D = no) T2: time (S: 1005 *±* 53 vs. PLA: 996 *±* 49, D = no)
Pinna et al. [43]	M, swimmers (*n* = 14)	34.7 *±* 7.5	Master athletes involved in regional and national competitions; training volume: average of 6.5 *±* 0.8 h per week; training frequency ranged 3–4 times/wk, with 3000–5000 m distance covered each time	42.7 *±* 2.6	BJ 500 mL (6 days)	5.5	3	Incremental swimming test	Workload (kg/min), VO_2_ (mL/min)	Workload (S: 6.7 *±* 1.1 vs. PLA: 6.35 *±* 1 D = yes), VO_2_ (S: 2741 *±* 454 vs. PLA: 2817 *±* 545, D = no)
Rokkedal et al. [44]	M, cyclists (*n* = 12)	29.1 *±* 7.7	Well-trained in performance level 4 [47,48]	66.4 *±* 5.3	BJ 140 mL (7 days)	12.4	2.75	Time trial 10 km	Power output (W), time (s), VO_2_ (mL/min)	Power output (S: 315.8 *±* 13.2 vs. PLA: 311.3 *±* 13.2, D = yes), time (S: 884,5 *±* 16 vs. PLA: 890,1 *±* 16, D = yes), VO_2_ (D = yes)
Shannon et al. [45]	M, runners and triathletes (*n* = 8)	28.3 *±* 5.8	Experience in competing in running events	62.3 *±* 8.1	BJ 140 mL (1 day)	12.5	3	Time trial 1500 m	Time (s), VO_2_ (mL/kg/min)	Time (S: 319.6 *±* 36.2 vs. PLA: 325.7 *±* 38.8, D = yes) VO_2_ (S: 53.4 *±* 6.8 vs. PLA: 53.9 *±* 6.9, D = no)
								Time trial 10 km	Time (s), VO_2_ (mL/kg/min)	Time (S: 2643.1 *±* 324,1 vs. PLA: 2649.9 *±* 319.8, D = no), VO_2_ (S: 49 *±* 6 vs. PLA: 48.6 *±* 6.3, D = no)
Wilkerson et al. [46]	M, cyclists (*n* = 8)	31 *±* 11	Well-trained subjects; training volume: 5.8 *±* 1.0 times per week and completed 11.1 ± 2.5 h training per week	63 *±* 8	BJ 500 mL (1 day)	6.2	2.5	Time trial 50 miles	Time (min), mean power output (W)	Time (S: 136.7 *±* 5.6 vs. PLA: 137.9 *±* 6.4, D = no), mean power output (S: 238 *±* 22 vs. PLA: 235 *±* 27, D = no)

M: male, F: female, VO_2max_: maximal oxygen uptake, VO_2peak_: peak oxygen uptake, BJ: beetroot juice, SmO_2_: vastus lateralis oxygen saturation, RPE: rating of perceived exertion, VCO_2_: expired carbon dioxide, D: statistical difference, T: trial, VT: ventilatory threshold, PLA: placebo, S: supplemented.

**Table 4 nutrients-12-01796-t004:** Summary table with studies on the cyclic endurance performance of trained athletes.

Sport	*n*	Sex	Number of Studies	Number of Significant Studies	Results on Respiratory and/or Performance Parameters	
					Performance	Respiratory
Cycling	101	All M	9	2 [35,44]	Time [35,44], MPO [35,44]	VO_2max_ [44]
Cycling and Triathlon	97	85 M, 12 F	6	2 [23,26]	Time [26], MPO [26]	VO_2peak_ [23] VO_2max_ [26]
Athletics and Triathlon	17	All M	2	1 [22]	Time, RPE	SmO_2_
Triathlon	26	12 M, 14 F	2			
Athletics	20	All M	2	1 [45]	Time	
Cross-country skiing	18	15 M, 3 F	2			
Rowing	10	All M	1	1 [33]	Time	
Swimming	34	24 M, 10 F	3	1 [43]	Time	Reduction of aerobic energy cost
Six different sports	323	Only one study exclusively with F [31] and four mixed studies [29,34,36,41]	27	8	7	5

M: male, F: female, VO_2max_: maximal oxygen uptake, VO_2peak_: peak oxygen uptake, SmO_2_: vastus lateralis oxygen saturation, RPE: rating of perceived exertion, MPO: mean power output.

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
