# Peer review of "Influence of Nitrate Supplementation on Endurance Cyclic Sports Performance: A Systematic Review"

_nutrients, 2020, doi:10.3390/nu12061796_

Round 1
Reviewer 1 Report
Overall comments:
The authors systematically reviewed the influence of nitrate supplementation on endurance cyclic sports performance. There are several major concerns that must be dealt with for this manuscript.
Firstly, the lack of novelty in this review. What is new compared to the review by McMahon et al, 2016; “The Effect of Dietary Nitrate Supplementation on Endurance Exercise Performance in Healthy Adults: A Systematic Review and Meta-Analysis.»? The references in the current study mostly overlap with the previous review.
Secondly, if the authors intend to focus on endurance cyclic sport, there are several limitations:
- You need to include a better definition of endurance cyclic sport already in the abstract for the reader to follow – now the rationale is very unclear
- Your aim to investigate the influence of nitrate supplementation on endurance cyclic sports performance. However, you do not answer this with the current study setup – the study selection criteria are broad (including all endurance sport, similar to the previous review by McMahon).
- Also, your writing is very unclear and it is difficult to get the message you try to deliver (see specific comments per section below)
Specific comments:
Abstract:
- Very unclear rationale – is your aim now to look at training level or effect of nitrate for endurance cyclic sport? You state; “Dietary nitrate supplementation produce an ergogenic effect by increasing exercise economy in recreationally active subjects. However, its effect on well-trained athletes remains unclear” – if this is a fact, why is there still a need to review the effect on endurance sport? Shouldn’t the review then focus on the training level? And if the relevance of power in cyclic sport is the rationale, why is this not emphasized
- “Statistically significant results were obtained in 35% of the studies” – you mean that nitrate supplementation improved some performance or physiological marker in 35% of the studies?
Introduction:
- Very unclear rationale and build up towards the aim
- Line 47 and out; you need to define cyclic sport much better, and connect this to the proposed effects of nitrate supplementation
- Your second aim “Also, was evaluated the main sources of dietetic nitrate, and dosage for ergogenic purposes” is both poorly written and explained, please adjust
Methods:
- Your search criteria do not account for the type of sports that you are interested in, seems like you just include all performance studies.
- Furthermore, even though you include all types of performance studies, your search ends up with fewer studies than the review by Mc Mahon et al. Partly the difference lies in the years included in your and their previous review. However, most references are similar and as such the question is “what is novel”?
Discussion:
- You state: “Meanwhile Bescós et al. [18], through the severe administration of dietetic nitrate in competitive cyclists and triathletes, figured out that even though there was not an improvement in the cardiorespiratory adaptation to low and medium exercise intensity, the consumption of oxygen was reduced at high intensity exercise. Jonvik et al. [22] stated that “plasma and the sprint performance repeatedly to the supplementation with beet juice don’t differ between recreational and competitive activities and elite sprint athletes”. This leads us to determine the importance of the type of exercise performed.” – if this is your main aim, this should be highlighted in the introduction – furthermore, your study selection criteria must be focused on this.
- The discussion section is not built up to answer your aim and lacks novelty
Author Response
Response to Reviewer 1 Comments
Point 1: The authors systematically reviewed the influence of nitrate supplementation on endurance cyclic sports performance. There are several major concerns that must be dealt with for this manuscript.
Firstly, the lack of novelty in this review. What is new compared to the review by McMahon et al, 2016; “The Effect of Dietary Nitrate Supplementation on Endurance Exercise Performance in Healthy Adults: A Systematic Review and Meta-Analysis.»? The references in the current study mostly overlap with the previous review.
Response 1: The big difference with respect to the review by McMahon et al. (2017) is about the criterion of the sample. McMahon did it with participants had to be healthy, human teens or adults and our sample is about trained athletes.
In addition, we focus exclusively on endurance and cyclic sports under normal training conditions. Therefore, conditions such as hypoxia, for example, and team sports are excluded. It is the big difference with the previous review.
Point 2: Secondly, if the authors intend to focus on endurance cyclic sport, there are several limitations:
You need to include a better definition of endurance cyclic sport already in the abstract for the reader to follow – now the rationale is very unclear
Response 2: Thank you very much for the clarification. Both in the summary and in the text has been changed and reword.
Point 3: Your aim to investigate the influence of nitrate supplementation on endurance cyclic sports performance. However, you do not answer this with the current study setup – the study selection criteria are broad (including all endurance sport, similar to the previous review by McMahon).
Response 3: As we have established in point 1, the main differences would be:
- Only well-trained athletes.
- In cyclic sports.
- Under normoxic conditions.
McMahon et al. (2017) there are all kinds of subjects, trained or amateurs. There are also hypoxic conditions, so therefore, there are things in common but there are many differences between the studies.
Finally, we have carried out a test of different methodological quality, PEDro vs McMaster Critical Review Form.
Point 4: Also, your writing is very unclear and it is difficult to get the message you try to deliver (see specific comments per section below).
Response 4: We apologize for any poor wording or misinterpretation. All your requests have been modified or information has been added. Please review the new document and hopefully it will be to your liking.
Point 5: Abstract: Very unclear rationale – is your aim now to look at training level or effect of nitrate for endurance cyclic sport? You state; “Dietary nitrate supplementation produce an ergogenic effect by increasing exercise economy in recreationally active subjects. However, its effect on well-trained athletes remains unclear” – if this is a fact, why is there still a need to review the effect on endurance sport? Shouldn’t the review then focus on the training level? And if the relevance of power in cyclic sport is the rationale, why is this not emphasized
Response 5: Thank you very much for the contribution. We have given you a new orientation based on your comments.
Point 6: Abstract: “Statistically significant results were obtained in 35% of the studies” – you mean that nitrate supplementation improved some performance or physiological marker in 35% of the studies?
Response 6: We were referring to% in the number of studies, but we have changed it to number and% so that it is better understood.
Point 7: Introduction: Very unclear rationale and build up towards the aim
Response 7: reword.
Point 8: Introduction: Also, Line 47 and out; you need to define cyclic sport much better, and connect this to the proposed effects of nitrate supplementation
Response 8: We have expanded this recommendation throughout the text. Hopefully you will like it more.
Point 9: Introduction: Your second aim “Also, was evaluated the main sources of dietetic nitrate, and dosage for ergogenic purposes” is both poorly written and explained, please adjust
Response 9: We have reviewed and expanded the studies again, you will see it in the summary tables that we have made with the major review.
Point 10: Methods: Your search criteria do not account for the type of sports that you are interested in, seems like you just include all performance studies. Furthermore, even though you include all types of performance studies, your search ends up with fewer studies than the review by Mc Mahon et al. Partly the difference lies in the years included in your and their previous review. However, most references are similar and as such the question is “what is novel”?
Response 10: As we have commented in point 1 and 3, we have tried to focus only on athletes trained in normoxic conditions to check the NO doses, protocols, administration of the last doses, and their results in this type of athletes and not others.
We have included all studies that athletes participated in national and international competitions.
McMahon et al. (2017) there were all kinds of subjects, trained or amateurs. There are also hypoxic conditions, so therefore, there are things in common but there are many differences between the studies.
Point 11: Discussion: You state: “Meanwhile Bescós et al. [18], through the severe administration of dietetic nitrate in competitive cyclists and triathletes, figured out that even though there was not an improvement in the cardiorespiratory adaptation to low and medium exercise intensity, the consumption of oxygen was reduced at high intensity exercise. Jonvik et al. [22] stated that “plasma and the sprint performance repeatedly to the supplementation with beet juice don’t differ between recreational and competitive activities and elite sprint athletes”. This leads us to determine the importance of the type of exercise performed.” – if this is your main aim, this should be highlighted in the introduction – furthermore, your study selection criteria must be focused on this.
Response 11: We have modified it. Thanks for the input.
Point 12: Discussion: The discussion section is not built up to answer your aim and lacks novelty
Response 12: Thank for the suggestion. We have clarified the arguments of the aim of study and built up the discussion including new data about the specific population (trained athletes).
Reviewer 2 Report
The novel aspect of this paper is the population- trained athletes. Since this is an important distinguishing characteristic, it would be good to learn more about the definition of trained athletes. A chart or graph that describes this in an easily readable format would be ideal. What type(s) of athletes? Sex? Age? VO2MAX- running, cycling, or otherwise? How is "well-trained" defined?
There are several sentences that need rewording. See attached document with comments.
Consider narrowing the population or breaking the results/discussion down further to sport specific, and/or sex, to further understand the role of nitrate.
Some edits would make a stronger article.
Author Response
Response to Reviewer 2 Comments
Point 1: The novel aspect of this paper is the population- trained athletes. Since this is an important distinguishing characteristic, it would be good to learn more about the definition of trained athletes. A chart or graph that describes this in an easily readable format would be ideal.
Response 1: Thank you for the recommendation, so we have carried it out. At first we did not do it because not all describe the experience or category of the sample.
Point 2: What type(s) of athletes? Sex? Age? VO2MAX- running, cycling, or otherwise? How is "well-trained" defined?
Response 2: These concepts are all collected in Table 2 as you will see. The same investigations were those that collected if the athletes were trained, well trained, elite.
Anyway, as you requested in point 1 and at the request of the other reviewer, the information on the athletes has been expanded.
Point 3: Consider narrowing the population or breaking the results/discussion down further to sport specific, and/or sex, to further understand the role of nitrate.
Response 3: Thank you very much for your consideration. We have made some modifications that we hope you like.
Point 4: There are several sentences that need rewording. See attached document with comments. Line 10: Confusing sentence. Endurance sports produce fatigue? And reduction of oxygen. Reword.
Response 4: Thank you very much for your consideration. We have made modifications on the abstract that we hope you like.
Point 5: Line 12: Endurance?
Response 5: Thanks, added.
Point 6: Line 19-21: Statistically significant results were obtained in 35% of the studies investigated, obtaining only one study with significant results in both respiratory parameters and performance measures
Response 6: We were referring to% in the number of studies, but we have changed it to number and% so that it is better understood. Reword.
Point 7: Line 33-35: Reword
Response 7: Reword. Supplementation with nitrate, either with pure nitrate salts or foods rich in nitrate, such as beet juice, has been proven to enhance sport performance of recreationally active young people in a positive way.
Point 8: Line 36-37: Supplementation with nitrate, either with pure nitrate salts or foods rich in nitrate, such as beet juice, has been proven to affect the sport performance of recreationally active young people in a positive way
Response 8: Reword. This ergogenic effect is mediated by an enhaced function of type II muscle fibres, a reduced ATP cost of muscle force production, the improvement of mitochondrial oxygen efficiency, through the reduction of the oxygen cost of exercise that increase vasodilation and blood flow to the skeletal muscle.
Point 9: Line 45: More and more athletes, not just recreational but professional too, are interested in consuming.
Response 9: More and more athletes, not just recreational but professional too, are interested in improve their performance so they use consuming this supplements like nitrates before there competition to have improvements in it.
Point 10: Line 47-48: Strategies that improve high intensity resistance performance are determinants for endurance cyclic sports
Response 10: it was a misprint. We wanted to say endurance.
Point 11: Line 53-54: The purpose of this study is realizing a systematic revision of scientific literature about the ergogenic effects that the supplementation of nitrate as a has in endurance cyclic athletes.
Response 11: We have further specified the study in relation to the aim of study and its suggestions.
Point 12: Page 12: How were "athletes" defined?
Response 12: As we have commented in point 2, the studies themselves set the guidelines for the level of athletes. There is no single criterion.
Thus, the studies mark criteria such as years of experience, amount of training per week, VO2max in minimum values for the inclusion criteria, ...
we have collected this data in the new table.
Point 13: Page 12: “studies used acute supplementation was taken into account, as well as others that carried out the intake of nitrate through several days”
Response 13: The average dose of nitrate was 8.7mmol/day (minimum 4mmol/day and 19.5mmol/day maximum). Acute and chronic supplementation protocols were found in the reviewed studies. 40.74% used chronic supplementation, 51.85% acute supplementation, using equally high and low doses for acute treatments as well as chronic, and 7.4% use both types of supplementation.
Point 14: Page 12: Would be nice to see a summary table of this information- differences in mmol of nitrate, timing of intake, duration, sport. etc.
Response 14: These values and others, are collected in Table 2 as you can see. If you think it appropriate to do it in another way, let us know.
Point 15: Page 12: What does this mean? “moderate to high level of cardiorespiratory fitness”
Response 15: What we mean is that athletes in all studies have moderate and high physiological values. Furthermore, we can describe and categorize the fitness of the subject group from the different investigations. These physiological variables may include peak power output (PPO), maximal aerobic capacity (VO2max), and submaximal characteristics such as anaerobic and ventilatory threshold.
Point 16: Page 12: Male vs female? Sport specific?
Response 16: male and cycling. Added.
Point 17: Page 13: Reword “NO has led a revolution in physiology and pharmacology research over the last decade affects to multiple biological mechanisms that influence sports performance such as smooth muscle relaxation which induces to vasodilation and thereby to and improved oxygen delivery and mitochondrial respiration”
Response 17: We have expanded and added more information in these paragraphs. Hopefully they are to your liking.
Point 18: Page 14: “The fact that none of the studies gave a significant effect using potassium nitrate or sodium nitrate as a nitrate-rich supplement may be because these supplementation forms were less effective”
Response 18: Other studies that have tried different forms of nitrate supplementation such as potassium nitrate or sodium nitrate did not obtain significant results. This would indicate that this form of supplementation may be less effective than beetroot juice. For example, a study conducted in 2016…..
Point 19: Page 15: How was this defined? “low and medium exercise intensity”
Response 19: The authors do not define it, they establish it thus: “The protocol of the test was divided into two parts: submaximal and maximal exercise intensity. Initially, the subjects completed four submaximal workloads corresponding to 2.0, 2.5, 3.0, and 3.5 W.kg/j of body mass with every load lasting for 6 min, interspersed with 3 min of passive recovery. Five minutes after completion of the submaximal workloads, subjects performed a continuous incremental exercise test to volitional exhaustion. Starting at 3.0 WIkgj1, the work rate increased by 0.5 W.kg/j every minute until task failure as a measure of exercise tolerance”.
During the tests, oxygen uptake (VO2), minute ventilation (VE), carbon dioxide production (VCO2), and the respiratory exchange ratio RER were measured breath-by-breath by a computerized gas analyzer (Cosmed Quark PFT-Ergo, Rome,Italy). Heart rate (HR) as well.
Point 20: Page 15: Too vague/broad “cycoergometer, with cyclists and triathletes, tests of 1 up to 80 km were made”
Response 20: We have expanded and added more information in these paragraphs. Hopefully they are to your liking.
Point 21: Page 15: This is confusing, and seems to say only the 10k distance has shown results? “Studies conducted with distances less than or greater than 10km did not obtained any significant improvement in exercise capacity”
Response 21: Thank you. We have modified the paragraph and linked it to different studies and distances, discussing it with other studies below.
Point 22: Page 16: Is this an endurance sport? “sufficient studies conducted in kayakers”
Response 22: In relation to your comments, we have carried out a new investigation check that we have modified. Now it corresponds more to what you establish and we have specified this section.
Point 23: Page 16: Are rowers and kayakers considerd the same?
Response 23: They are not the same, not at all. The Olympic rowing distances are 2000, for the kayak they can be 200, 500 and 1,000 meters.
Rowers are considered endurance athletes, specifically this study by Hoon et al. [26] use the keyword "endurance" and quotes: Competitive 2000-m rowing typically requires continuous whole-body work at ~ 90% of VO2max for approximately 6 to 7 minutes "using their sample as endurance athletes.
Point 24: Page 16: Who were these athletes? Well-trained? In what sport? How trained?
Response 24: The study by Pinna et al. [25] with moderately trained male swimmers is a clear example of improving exercise efficiency. The training frequency of swimmers ranged from 3 to 4 times/week, with 3.000–5.000 m distance covered each workout. With a supplementation of 5.5 mmol of nitrate for 6 days an incremental test was carried out until exhaustion in the pool, using an elastic cord, connected to a digital dynamometer tied with a belt to the subject for workloads. The results showed that through the intake of beet juice a greater workload and a lower aerobic energy consumption were carried out, together with the same VO2.
Point 25: Page 16: Add definition of well-trained (VO2MAX?, years of training? etc)
Response 25: This definition and information has been included in table X, so whenever we speak it will be considered with the exposed data.
Point 26: Page 17: What does this mean? Are they really non-responsers or is this methodological? “athletes who benefit more than others, and there may be responders and non-responders”
Response 26: It is a point about the study because Bescós et al. [17] classified them thus methodologically. We add the word "methodologically".
Point 27: Page 17: Reword “The improvement of sports performance through the decrease in oxygen consumption is a characteristic that is not so clear in high performance”
Response 27: we wanted to say "uptake" vs “consumption”.
Point 28: Page 17: How much nitrate, range?
Response 28: The variety of results makes it difficult to draw clear conclusions about nitrate supplementation on endurance cyclic sports performance. Although most of the studies showed no significant differences on the performance of the endurance trained athletes with nitrate supplementation, some of them indicated that with the same maximal and mean power, the VO2 values decrease after a nitrate supplementation. Furthermore, the time to exhaustion was delayed with nitrate supplementation in races time trial between 5 and 30 minutes. Both results might enhance the performance.
As for the dose necessary for a significant effect, it remains unclear, since some results with acute doses of 12mmol of nitrate and others with smaller doses (up to 6mmol / day of nitrate). Finally, the best moment for the ingestion is between 2-3 hour before the competition.
Further investigations are required to confirm the findings of the present study.
Round 2
Reviewer 1 Report
The authors systematically reviewed the influence of nitrate supplementation on endurance cyclic sports performance. The authors have done a good job responding to most of the comments, however there are still a few open concerns before this manuscript to be considered for publication. I still think the manuscript is not written with a clear message to the reader and especially the introduction and discussion. There is also a need for improvement of the English language and I suggest getting help from a native speaker for this. Furthermore, now it is very difficult to read from your cover letter where you have made the changes per comments. In a following revision, please include line numbers or the specific change in text within the cover letter.
- I still question the novelty of this review. Even though the McMahon study includes healthy subjects, the focus is on endurance sport performance. Therefore, in order to show the novelty of the current study, more emphasize and a clearer discussion must be included for the parameters you evaluate here: a) (highly) trained athletes (and whether training level is an important factor), b) cyclic sports (which sports are included, why you focus on cyclic sports and how do you compare the effects to other types of sports), c) why is this review on athletes so different to the review on healthy subjects who are performing endurance sports (McMahon review). I think there should be a clear message about this to the reader in the introduction of the manuscript.
- You now clearify the definition cyclic sport – but there is no rationale to why you choose this, as opposed to e.g. intermittent sport (where several studies are performed and ergogenic effects of nitrate are found). Please include a clear rationale to your choice of aim and target group.
- Response to previous point 6: “Response 6: We were referring to% in the number of studies, but we have changed it to number and% so that it is better understood.”
- Why have you now changed to 25.7%? Was 35% not correct?
- Is the study that found significant results on both respiratory and performance measures included in this number?
- This sentence is still not correct English – please improve
- I still think the rationale and build up towards the aim is not clarified (as described above) – please improve the introduction
- Your second aim “Also, was evaluated the main sources of dietetic nitrate, and dosage for ergogenic purposes..” is still not correct English – and also not well explained in the introduction – you need a rationale towards this aim. Please include this.
- You still not have a clear description on how you extracted the cyclic sport studies based on your very wide search criteria. Please include a better description of the factors of eligibility.
- You still have not answered the point regarding the sentence “Bescós et al. [18], through the severe administration of dietetic nitrate in competitive cyclists and triathletes, figured out that even though there was not an improvement in the cardiorespiratory adaptation to low and medium exercise intensity…”. Even though you have adapted the sentence, if this is an aim of the study, you need to clearify in your study selection criteria and highlight in the introduction that the intensity of exercise is a parameter for the importance of the type of exercise. In that case, your discussion should also be clearer in the statement of the importance of exercise intensity.
- I agree with the other reviewer that narrowing the population or breaking the results/discussion down further to sport specific, and/or sex, to further understand the role of nitrate would improve the manuscript. However, I do not see sufficient modifications on this point in the discussion.
Author Response
Dear Reviewer,
Due to some problems with the management of the payment for the English translation and for personal reasons, we have delayed a few days more than expected.
We are sorry for the delay and hopefully this time it is fit for your approval.
Best.
Response to Reviewer 1 Comments
Point 0. The authors systematically reviewed the influence of nitrate supplementation on endurance cyclic sports performance. The authors have done a good job responding to most of the comments, however there are still a few open concerns before this manuscript to be considered for publication. I still think the manuscript is not written with a clear message to the reader and especially the introduction and discussion. There is also a need for improvement of the English language and I suggest getting help from a native speaker for this. Furthermore, now it is very difficult to read from your cover letter where you have made the changes per comments. In a following revision, please include line numbers or the specific change in text within the cover letter.
Response 0: Thanks for the suggestions. We have improved the introduction and the discussion, as well as we have sent the document to the translators of the MDPI who have given us the ok, for this reason that it took us several days to return the document. We really apologize because the numbers were deleted. Again, with the modifications, it is difficult to specify your previous recommendation in the main text with the current page line.
Point 1. I still question the novelty of this review. Even though the McMahon study includes healthy subjects, the focus is on endurance sport performance. Therefore, in order to show the novelty of the current study, more emphasize and a clearer discussion must be included for the parameters you evaluate here: a) (highly) trained athletes (and whether training level is an important factor), b) cyclic sports (which sports are included, why you focus on cyclic sports and how do you compare the effects to other types of sports), c) why is this review on athletes so different to the review on healthy subjects who are performing endurance sports (McMahon review). I think there should be a clear message about this to the reader in the introduction of the manuscript.
Response 1:
- a) Our study objective is to reduce NO supplementation in trained athletes with respect to what the rest of the studies that mix different environments and populations. We only focus on Endurance Cyclic Sports Performance and their contributions.
In relation to the sample and its definition of trained, well-trained or elite. It is the authors themselves who highlight it and we point out those characteristics. But we have not defined this sample.
- b) As before, we are only interested in cyclic sports in the review, not other sports, moreover, in the largest review we carry out at your suggestion, we eliminate some of the articles that, being cyclic, they were not endurance, for focus even more on the aim of study as you requested. Thanks for the contribution.
- c) Our research is inspired by McMahon's study, which states that “Further work is needed to understand the optimal dosing strategies, which population is most likely to benefit, and under which conditions dietary nitrates are likely to be most effective for performance”.
Hence, we have done it exclusively in this type of sample and conditions. Hence, we have done it exclusively in this type of sample and conditions.
Point 2. You now clearify the definition cyclic sport – but there is no rationale to why you choose this, as opposed to e.g. intermittent sport (where several studies are performed and ergogenic effects of nitrate are found). Please include a clear rationale to your choice of aim and target group.
Response 2: We understand that it is a great possibility, but we have preferred to focus only on one type of sample. Perhaps another investigation or another review could be to compare the different sports, or to include each sport according to their nature and metabolic requirements, but it is not our purpose of the study, which is exclusively to focus on Endurance Cyclic Sports Performance and their contributions.
Point 3. Response to previous point 6: “Response 6: We were referring to% in the number of studies, but we have changed it to number and% so that it is better understood.”
- Why have you now changed to 25.7%? Was 35% not correct?
Based on the second review and the modifications and contributions made by the reviewers, the number of articles included in the document and the number of significant studies finally changed. So it was a mistake.
The review consists of 27 articles, of which 8 (29.63%) are significant.
- Is the study that found significant results on both respiratory and performance measures included in this number?
Not all of them,
We have improved and expanded Table 4 for your understanding.
As well as during the text, mainly in the current lines 200-211 (3.4. Grouped and sub-groups) together with the information in table 3 (main table) it is much clearer.
- This sentence is still not correct English – please improve Thank you. Corrected. The phrase has been modified and restructured and has the ok of native speaker. We attach the certificate and hopefully it is to your liking.
Point 4. I still think the rationale and build up towards the aim is not clarified (as described above) – please improve the introduction.
Response 4: We have tried to improve it again. Hopefully you like it.
Point 5. Your second aim “Also, was evaluated the main sources of dietetic nitrate, and dosage for ergogenic purposes..” is still not correct English – and also not well explained in the introduction – you need a rationale towards this aim. Please include this.
Response 5: Thank you. We have extensively modified the introduction. Corrected. Also we have restructured and it has the ok of native speaker. We attach the certificate and hopefully it is to your liking.
Point 6. You still not have a clear description on how you extracted the cyclic sport studies based on your very wide search criteria. Please include a better description of the factors of eligibility.
Response 6: We have included one more point in the inclusion criteria and defined the eligibility criteria throughout the text. Although we understand that cyclical sport is one that repeats the same gestures in the same way repeatedly marking frequency values as highlighted by literature. Furthermore, it is such an established criterion that a great many modern literature do not even provide evidence or justify it with other authors.
Point 7. You still have not answered the point regarding the sentence “Bescós et al. [18], through the severe administration of dietetic nitrate in competitive cyclists and triathletes, figured out that even though there was not an improvement in the cardiorespiratory adaptation to low and medium exercise intensity…”. Even though you have adapted the sentence, if this is an aim of the study, you need to clearify in your study selection criteria and highlight in the introduction that the intensity of exercise is a parameter for the importance of the type of exercise. In that case, your discussion should also be clearer in the statement of the importance of exercise intensity.
Response 7: Thanks for the comment.
We have introduced the term “intensity” inside the inclusion criteria:
“(I) depicting a well-designed experiment that included the ingestion of a dose of nitrate, or nitric oxide or beetroot juice before and/or during different intensity exercise in healthy trained athletes;”
We have modified the introduction section giving more importance to the type of intensity in these kind of sports and its relation with success in the specific sample of the study.
- “(anaerobic and aerobic way),”
- “But why its important to work on strategies that improve high intensity endurance in cyclic sports? In these sports, which involve repetitive movements and prevalent aerobic charge, the expended effort typically lasts longer than five minutes, primarily depending on the metabolic level of the oxidative processes involved; so the intensity use to be lower in relation to the volume. However, in cyclic athletes, one of the key factors for succeed is the capacity of perform at the end of the competition at higher level than the rivals; and in that situation, high intensity efforts could be determinant for the victory. When high intensity effort occurs, there are physiological factors that limit performance”
In the discussion section and mainly in relation to the significant studies, we have clarified the role of intensity in relation to the positive results obtained in performance and / or respiratory parameters after nitrate intake
- “Balsalobre et al. [19], after supplementing the runners with a dose of 6.5mmol/day of nitrate for 15 days, performed an incremental test until exhaustion on treadmill (from low to high intensity), with significant results in time (S: 1269 ± 53.6 vs. PLA: 1230 ± 73.5s).; which means that supplemented athletes performed better. The highest intense part of the test is used to determine the speed associated with the VO2max.”.
- “the consumption of oxygen (Vo2max) was reduced at high intensity exercise being much more efficient. This leads us to determine the importance of the high intensity in the performance of the cyclic sports.”
- “which means in fact that they performed at a higher intensity.”
- “These results would show that as distances are greater, the importance of intensity as a determining factor in performance decreases.”
- “well-trained”
- “athletes”
Point 8. I agree with the other reviewer that narrowing the population or breaking the results/discussion down further to sport specific, and/or sex, to further understand the role of nitrate would improve the manuscript. However, I do not see sufficient modifications on this point in the discussion.
Response 8: With the new description in Table 4 and the discussion modified, we think we have improved this suggestion.
Thank you very much for your contributions to improve the document.